# XTalker: Turn, Smile, and Speak in Controllable Talking Portrait Animation

## Abstract

Audio-driven portrait animation enables lifelike talking faces from a static image and audio input, and has become a key task for creating realistic digital humans. Existing methods mainly follow either latent- or parameter-based paradigms, yet they either suffer from limited controllability, high computational cost, or lack broader expressivity (emotion and head diversity) beyond lip synchronization. To address these limitations, we present qualitative and quantitative analyses of parameter representations, focusing on the disentanglement of head keypoints, and discover that facial representation can be decomposed into three interpretable subspaces: lip-phoneme synchronization governed by audio envelope dynamics, emotion modulation conditioned on semantic features and reference labels, and head motion controlled by user-defined curves. Building on this insight, we propose XTalker, a fast and controllable generation framework based on flow matching. It employs a unified MM-DiT backbone to jointly encode portrait and audio signals, followed by three lightweight heads: a talking head for accurate lip movement driven by audio envelope, an emotion head for synthesizing facial expressions from emotion labels, and a pose head for animating diverse head trajectories. Extensive experiments show that XTalker runs in real-time and achieves competitive lip-sync accuracy while improving emotion and motion expressivity, making it well-suited for controllable portrait animation applications.

## 1 Introduction

Audio-driven portrait animation (Siarohin et al., 2019; Chen et al., 2019; Prajwal et al., 2020), which aims to synthesize photorealistic talking portraits from a single reference image and speech audio, plays a critical role in enabling realistic digital humans for applications such as multi-modal agents, virtual avatars, and low-shot dubbing. The central criterion is **expressivity**, defined as the ability to produce temporally coherent and semantically rich portraits from audio alone, spanning several dimensions: lip-audio synchronization, emotion diversity, and head motion controllability (Toshpulatov et al., 2023). Achieving these dimensions requires abundant high-quality and fine-grained annotations from raw videos. In the absence of such annotations, it is challenging to build expressive models under limited supervision and without auxiliary driving signals (e.g., reference videos), underscoring the need for interpretable representations to support expressivity in low-data settings.

Existing methods for audio-driven portrait animation largely fall into two categories: latent and parameter representations. Latent-based models (Xu et al., 2024a; Cui et al., 2024a;b; 2025; Ki et al., 2024; Cheng et al., 2024; Ji et al., 2024) operate in spatial latent spaces and employ powerful generative backbones with diffusion techniques (Ho et al., 2020) to synthesize video from audio. While they achieve realistic results, they require large-scale paired datasets, incur high computation and lack semantic interpretability for fine-grained controllability. Parameter-based methods (Zhang et al., 2021a; Wei et al., 2024; Guo et al., 2024; Xu et al., 2024b; Cao et al., 2024) instead utilize structured, low-dimensional representations such as facial landmarks (Blanz & Vetter, 2023; Wang et al., 2019; 2021). Their representations are compact and interpretable, suitable for low-data expressive modeling. However, most parameterized systems mainly focus on lip synchronization, whereas fine-grained control over emotion diversity and head motion remains underexplored.

To address these issues, we revisit parameter representations with a focus on disentangling interpretable facial subspaces. Through ablation studies and quantitative analyses on the Face-Vid2Vid

series (Wang et al., 2019; 2021; Guo et al., 2024), we find that facial motion can be systematically decomposed into three coordinated components: 1) lip-phoneme synchronization, governed by the temporal dynamics of the audio envelope; 2) emotional diversity, modulated through mapped parameter features; and 3) head motion controllability, guided by user-defined motion curves. This structured decomposition facilitates more precise control, stronger expressivity, and better generalization, forming a robust foundation for a unified and lightweight framework for expressive generation in low-data regimes.

Building on this insight, we propose XTalker, a fast and controllable framework for expressive audio-driven portrait animation. XTalker adopts a flow-matching generation paradigm built upon a unified MM-DiT backbone that encodes global features from both portrait and audio signals. On top of this, three lightweight heads are introduced: 1) The emotion head conditions on a given label to modulate expression keypoints, enabling the synthesis of distinguishable facial expressions across diverse basic emotions; 2) The talking head extracts temporal amplitude features from the audio envelope to generate temporally coherent and accurate lip-phoneme synchronization; and 3) The pose head maps user-defined motion curves to dynamic head trajectories, allowing explicit control over head motion variations and speaking styles. With systematic optimization strategies, XTalker runs at 28.21 FPS on a single RTX 4090, while achieving competitive lip-sync accuracy and superior emotion and motion expressivity compared with prior works, highlighting its strong potential for real-world deployment in controllable generation scenarios.

The main contributions are summarized as follows.

- We decompose facial dynamics into three interpretable subspaces: lip sync, emotion, and head motion, offering a structured view of parameter-based animation.
- We design XTalker, a lightweight framework with three specialized branches, each addressing a specific modality for fine-grained control and expressivity.
- Our model achieves real-time performance and superior experimental results in lip-sync, emotion, and pose diversity, outperforming prior controllable methods.

## 2 RELATED WORK

Audio-driven portrait animation aims to generate realistic talking-head videos from a reference portrait image and input audio, benefiting from recent advances in generative modeling with Diffusion (Sohl-Dickstein et al., 2015; Song & Ermon, 2019; Ho et al., 2020; Song et al., 2020) and Flow-matching (Lipman et al., 2022; Liu et al., 2022) techniques. Early methods regressed facial parameters and rendered outputs via traditional graphics pipelines, followed by a clear evolution: from parameterized graphics pipelines (Fan et al., 2015; Pham et al., 2017; Tzirakis et al., 2020), to GAN-based neural rendering (KR et al., 2019; Prajwal et al., 2020; Guan et al., 2023; Goyal et al., 2023), and now to large-scale generative models based on latent or parameter representations.

**Latent Representation.** Recent diffusion-based models (Chung et al., 2017; Karras et al., 2017; Liu et al., 2024; Tian et al., 2024; Ki et al., 2024; Chen et al., 2025; Xu et al., 2024a; Cui et al., 2024a;b; 2025; Li et al., 2024; Fan et al., 2025; Ji et al., 2024) enhance expressivity by incorporating conditioning signals such as motion trajectories or facial landmarks. Operating in spatial latent spaces with diffusion or flow-matching, they produce realistic results but demand large paired datasets. For example, Sonic (Ji et al., 2024) achieves robust lip–audio synchronization but relies on carefully aligned audiovisual data, while Float (Ki et al., 2024) improves emotional fidelity through landmark-conditioned diffusion yet offers limited controllability across identities.

**Parameter Representation.** Recent methods project facial motion into compact nonlinear parameter spaces, enabling expressive animation (Thies et al., 2016; Suwajanakorn et al., 2017; Kim et al., 2018; Siarohin et al., 2019; Wang et al., 2021; Zhang et al., 2023). Broadly, these works fall into two categories: explicitly supervised approaches that rely on labeled geometry or paired annotations (Kumar et al., 2017; Kim et al., 2018; Zhang et al., 2023; Wei et al., 2024) and implicit or generative approaches that avoid explicit 3D supervision by exploiting self-/unsupervised learning or generative priors (Thies et al., 2016; Siarohin et al., 2019; Wang et al., 2021; Ma et al., 2023; Guo et al., 2024). Built on LivePortrait (Guo et al., 2024), VASA-1 (Xu et al., 2024b) and JoyVASA (Cao et al., 2024) extend parameterized representations to large-scale dataset for vivid synthesis, highlighting parameter spaces as a promising balance of compactness.

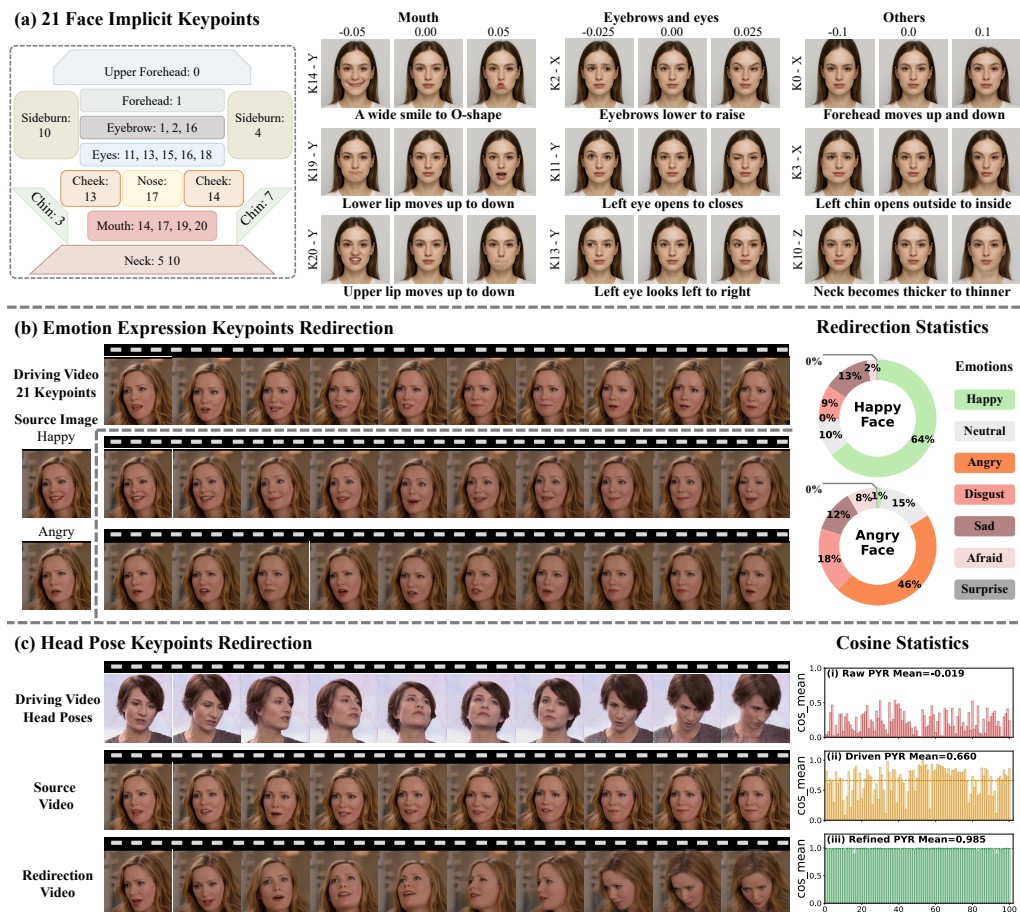

Figure 1: Analysis of Parameter Representation. (a) Impacts of implicit keypoint disentanglement on portrait controllability, particularly in the mouth and eye regions (all keypoints in Table 3). K14-Y denotes the Y-axis of keypoint 14; (b) Emotion keypoints redirection enables the generation to inherit expressions from an emotional reference image. In our evaluation on 100 test samples, happy and angry references preserve emotion in 64% and 46% of the generated frames, respectively; and (c) Head pose redirection enables the generated video to inherit diverse head motion patterns from the driving video. In a cosine similarity evaluation across 100 test cases, we observe a consistent improvement in head motion alignment between the generated and driving sequences.

## 3 METHODOLOGY

### 3.1 PROBLEM STATEMENT

**Preliminary.** Unlike latent-based approaches such as VAE-style image or video generation that operate directly in high-dimensional pixel space, parameterized methods disentangle appearance and motion into a compact, low-dimensional representation. A representative paradigm is the the Face-Vid2Vid family (Wang et al., 2019; 2021), further advanced by *LivePortrait* (Guo et al., 2024), which enables one-shot animation of a portrait image $\mathbf{I}_s$ driven by a $N$-frame video sequence $\{\mathbf{I}_1, \mathbf{I}_2, \dots, \mathbf{I}_N\}$. Formally, an appearance encoder $\mathcal{F}_{\text{app}}$ extracts a 3D feature volume $\mathbf{f}_s \in \mathbb{R}^{H \times W \times D \times C}$ from $\mathbf{I}_s$, while a motion encoder $\mathcal{M}_{\text{mot}}$ estimates parameters $(s, \mathbf{R}, \mathbf{o}, \boldsymbol{\delta})$ with canonical keypoints $\mathbf{X}^c \in \mathbb{R}^{K \times 3}$ ($K = 21$), where $s \in \mathbb{R}$ indicates scale coefficient, $\mathbf{R} \in \mathbb{R}^{3 \times 3}$ head pose coefficient, $\mathbf{o} \in \mathbb{R}^3$ displacement coefficient, and $\boldsymbol{\delta} \in \mathbb{R}^{K \times 3}$ residual expression keypoints. The transformed implicit keypoints from the source and driving frames are computed as

$$\mathbf{X}_s = s_s(\mathbf{X}_s^c \mathbf{R}_s + \boldsymbol{\delta}_s) + \mathbf{o}_s, \quad \mathbf{X}_{d_i} = s_{d_i}(\mathbf{X}_{d_i}^c \mathbf{R}_{d_i} + \boldsymbol{\delta}_{d_i}) + \mathbf{o}_{d_i}. \quad (1)$$

To drive the reference portrait, $\mathbf{X}_s$ interacts with driven $\mathbf{X}_{d_i}$ to form a blended representation $\hat{\mathbf{X}}_{d_i}$. A dense deformation field $\mathbf{DF} \in \mathbb{R}^{H \times W \times D \times 3}$ is then predicted by $\mathcal{W}_{def}$ network based on the correspondence between $\mathbf{X}_s$ and $\hat{\mathbf{X}}_{d_i}$, representing the 3D spatial displacement at each voxel. The warping function applies the field $\mathbf{DF}$ to the source features, producing a motion-aligned representation that is subsequently decoded by $\mathcal{G}_{\text{spade}}$ to produce the animated frame as

$$\hat{I}_{d_i} = \mathcal{G}_{\text{spade}}\big(\mathbf{Warp}(\mathbf{f}_s, \mathcal{W}_{def}(\mathbf{X}_s, \hat{\mathbf{X}}_{d_i})), \quad \hat{\mathbf{X}}_{d_i} = s_{d_i}(\mathbf{X}_s^c \mathbf{R}_{d_i} + (\boldsymbol{\delta}_s + \boldsymbol{\delta}_{d_i} - \boldsymbol{\delta}_{d_0})) + \mathbf{o}_{d_i}. \quad (2)$$

By disentangling appearance features from interpretable motion parameters, video-driven LivePortrait achieves efficient and controllable animation. In contrast, controllability tends to be absent in audio-driven diffusion-based frameworks such as VASA-1 (Xu et al., 2024b) and JoyVASA (Cao et al., 2024), where expressivity remains implicit and difficult to modulate in a unified structure.

**Representation Disentanglement.** Despite progress in achieving more controllable animation through implicit or non-visible keypoints in video-driven generation, there is still limited analysis of the functional roles of these keypoints, particularly their contribution to the overall expressivity of portrait animation. To bridge this gap, we revisit parameter representations with an emphasis on disentangling interpretable facial subspaces. Specifically, we perform a controlled linear traversal over each facial keypoint to assess its individual influence on portrait deformation, visualized in Figure 1(a). The results reveal a clear disentanglement structure among the keypoints, with those in the mouth region identified as the primary contributors to lip-phoneme synchronization. Building on this observation, we demonstrate that the identified disentangled keypoints can serve as effective control handles for modulating portraits across three complementary expressivity axes:

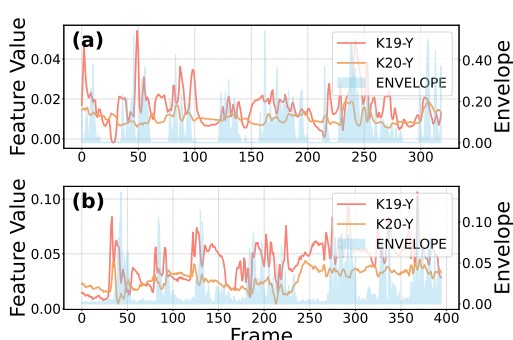

Figure 2: Correlation between lip and audio envelope. Mouth related keypoints exhibit high correlation with the audio envelope in raw video samples. More samples can be found in **Appendix C.2**

1) **Lip-phoneme Synchronization**: In audio processing, the amplitude envelope is strongly correlated with the underlying signal, capturing temporal rhythm and energy. Yet this property has not fully been explored or leveraged. We analyze its synchronization with mouth keypoints and find that keypoint trajectories closely follow the envelope, as shown in Figure 2, yielding precise lip-phoneme synchronization and confirming envelope dynamics as a reliable basis for coherent articulation.

2) **Emotional Diversity**: Emotion annotation on a per-frame basis is typically impractical for training portrait animation. We explore whether parameterized representations can preserve expression by modifying only a single reference frame. As shown in Figure 1(b), redirecting emotion-related keypoints from an affective image enables the generated sequence to retain that emotion. In our evaluation on 100 test samples, happy and angry references preserve the target emotion in 64% and 46% of frames, respectively, confirming the effectiveness of this approach. Although such references can be synthesized using text-to-image models (e.g., nano-banana), a unified framework with built-in emotion control is ultimately more desirable.

3) **Head Motion Controllability**: Controlling head motion in a precise and diverse manner remains a challenge in audio-driven generation. We examine whether parameterized representations allow effective control by redirecting pose-related keypoints from the driving video. As shown in Figure 1(c), this redirection enables the generated sequence to inherit head motion patterns. In a cosine similarity evaluation across 100 test cases, we observe consistent improvement in alignment with the driving sequence. To further enhance controllability, we propose guiding head motion using user-defined curves, enabling flexible and intuitive trajectory design.

These findings confirm that interpretable parameter subspaces not only support expressive decomposition, but also provide a viable basis for designing lightweight and controllable animation systems.

Figure 3: Overview of the XTalker framework. (a) The overall structure adopts a flow-matching generation paradigm with a unified MM-DiT backbone, which jointly encodes global representations from the source image and source audio, followed by three lightweight heads designed for emotion modulation, lip-phoneme synchronization, and head pose controllability; (b) A pretrained Transformer converts the source image into emotion-conditioned counterparts, serving as supervisory signals for the emotion head; and (c) An LLM-guided curve-pose synthesis translates predefined motion curves into head motion trajectories, providing ground-truth supervision for the pose head.

## 3.2 PROPOSED XTALKER

Motivated by our findings on facial behavior disentanglement, we design **XTalker** to exploit interpretable parameter subspaces for controllable audio-driven portrait animation. Given a source image $\mathbf{I}_s$ and driving audio $\mathbf{A}$, we introduce three control modalities: a discrete emotion label $l_{\text{emo}}$ encoded as an $L$-dimensional one-hot vector, an amplitude envelope $\mathbf{env} \in \mathbb{R}^N$ capturing audio rhythm, and a user-defined trajectory $\mathbf{Cur} \in \mathbb{R}^N$ for head motion. These serve as semantic priors and are embedded as conditioning vectors through lightweight control heads, as described below.

**Multi-modal DiT Backbone.** We first extract residual expression keypoints $\boldsymbol{\delta}_s \in \mathbb{R}^{1 \times (K \times 3)}$ from the source image $\mathbf{I}_s$. An expression noise $\boldsymbol{\epsilon} \in \mathbb{R}^{N \times (U+K \times 3)}$ is initialized, representing the $U$ head pose coefficient and $K \times 3$ expression keypoints. In parallel, the driving audio $\mathbf{A}$ is encoded using a pretrained Wav2Vec model (Baevski et al., 2020) to obtain semantic embeddings $\hat{\mathbf{A}} \in \mathbb{R}^{N \times 768}$. Both modalities are projected into a unified embedding space of dimension $d_c$, and concatenated along the token axis before being passed to the DiT backbone $\mathcal{B}_{\text{dit}}$:

$$\mathbf{H}_{in} = \mathcal{B}_{\text{dit}}([\mathbf{MLP}(\boldsymbol{\delta}_s); \mathbf{MLP}(\boldsymbol{\epsilon}); \mathbf{MLP}(\hat{\mathbf{A}})] \in \mathbb{R}^{(1+2N) \times d_c}, \tag{3}$$

where $[;]$ represents the concat operation, and $\mathbf{H}_{in}$ is the intermediate latent representation capturing both source emotion keypoint representation and audio embeddings.

**Disentangled Prediction Heads.** To enable disentangled control for animation expressivity, we introduce three lightweight heads that predict semantically grounded velocity fields, each modulated by a distinct prior. The emotion head leverages a discrete label $l_{\text{emo}}$ to impose a global affective bias, capturing long-term expression attributes such as smile intensity or eyebrow movement in the source frame. The talking head incorporates the amplitude envelope $\mathbf{env} \in \mathbb{R}^N$ to model local lip dynamics aligned with the speech rhythm. The pose head is conditioned on a user-defined trajectory $\mathbf{Cur} \in \mathbb{R}^N$, which specifies desired head movement patterns over time. These priors are projected to $d_c$ dimension and fused with $\mathbf{H}_{in}$ to produce distinct velocity fields:

$$\mathbf{v}_{\text{e}} = \mathcal{H}_{\text{e}}([\mathbf{H}_{in}, \mathbf{MLP}(l_{\text{emo}})]), \ \mathbf{v}_{\text{t}} = \mathcal{H}_{\text{t}}([\mathbf{H}_{in}, \mathbf{MLP}(\mathbf{env})]), \ \mathbf{v}_{\text{p}} = \mathcal{H}_{\text{p}}([\mathbf{H}_{in}, \mathbf{MLP}(\mathbf{Cur})). \tag{4}$$

**Flow-Matching Inference.** Each predicted velocity field $\mathbf{v}_{\text{e}}, \mathbf{v}_{\text{t}}, \mathbf{v}_{\text{p}}$ is integrated over time and applied to expression keypoints $\boldsymbol{\delta}_s$ and expression noise $\boldsymbol{\epsilon}$, producing three expressive representations:

$$\hat{\boldsymbol{\delta}}_s = \boldsymbol{\delta}_s + \int_0^1 \mathbf{v}_{\text{e}}(t) \, dt, \quad \hat{\boldsymbol{\epsilon}} = \boldsymbol{\epsilon} + \int_0^1 [\mathbf{v}_{\text{t}}(t); \mathbf{v}_{\text{p}}(t)] \, dt. \tag{5}$$

Then we decompose $\hat{\epsilon}$ into head motion $\hat{\mathbf{R}}_{d_i}$ and expression keypoints $\hat{\boldsymbol{\delta}}_{d_i}$. Following Eqs. (1) and (6), we decode them into residual emotion keypoints and apply a dense warping function to the source image $\mathbf{I}_s$, with unchanged scale and displacement coefficients to produce the final video frames:

$$\hat{I}_{d_i} = \mathcal{G}_{\text{spade}}\big(\mathbf{Warp}(\mathbf{f}_s, \mathcal{W}_{def}(\mathbf{X}_s, \hat{\mathbf{X}}_{d_i}))\big), \quad \hat{\mathbf{X}}_{d_i} = s_s(\mathbf{X}_s^c(\gamma\hat{\mathbf{R}}_{d_i}) + (\alpha\hat{\boldsymbol{\delta}}_s + (1-\alpha)\hat{\boldsymbol{\delta}}_{d_i})) + \mathbf{o}_s, \quad (6)$$

where $\gamma$ is a constant (default 1) that acts as a scaling factor to adjust the head motion. Importantly, the foregoing process reflects our inference-time reasoning rather than available supervision. During training, learning the emotion head and pose head is nontrivial: standard video datasets rarely provide aligned emotion pairs, nor do they offer user-defined head motion trajectory as ground truth. To bridge this supervision gap, *Emotion and Curve-Guided Pose Conditioning* is introduced to construct targets for $l_{\text{emo}}$ and derive curve-pose priors to guide the disentangled velocity fields.

## 3.3 EMOTION AND CURVE-GUIDED POSE CONDITIONING

**Emotion Expression Transformer.** To disentangle emotional expression from pose and audio, we introduce a lightweight *Emotion Expression Transformer* that conditions on emotion labels to regress target embeddings in keypoint space. Given the source embedding $\boldsymbol{\delta}_s \in \mathbb{R}^{K \times 3}$ and a one-hot emotion vector $l_{\text{emo}}$, a DiT-style regressor predicts the emotion-modulated embedding: $\boldsymbol{\delta}_e = G_\phi([\boldsymbol{\delta}_s; l_{\text{emo}}]) \in \mathbb{R}^{K \times 3}$. This embedding is fused with appearance features via the warping module and SPADE decoder to guide expressive synthesis. Further details are presented in **Appendix D.1**.

**LLM-Guided Curve-Pose Synthesis.** To enhance head motion controllability, we design a *curve-pose branch* that uses an LLM to generate predefined curves mapped into Pitch–Yaw–Roll (PYR) increments through a diagonal transform. For a normalized curve trajectory $\mathbf{Cur} \in \mathbb{R}^N$, we compute the head motion $\mathbf{R}_{d_i}$ mapping by diagonal transform. Although the diagonal transform provides a direct mapping from predefined curves to PYR increments, relying solely on this deterministic mapping is insufficient: curve priors vary significantly across prompts and users, and head motion must remain consistent with emotion and lip dynamics. We thus employ a learned curve–pose branch that adaptively normalizes and fuses diverse curve priors, enabling robust and generalizable head motion control beyond handcrafted mappings. Further details are given in **Appendix E.1**.

## 3.4 OVERALL OPTIMIZATION

**Noise Initialization.** To guide early training toward smooth and semantically meaningful motion, we initialize the expression noise $\boldsymbol{\epsilon} \in \mathbb{R}^{N \times (U + K \times 3)}$ by combining temporal interpolation, envelope-driven modulation, and selective jitter (based on analysis in **Appendix C**). We first sample $(N/N_{\text{seg}} + 1)$ Gaussian anchors and interpolate them to obtain a temporally smooth sequence with $N$. Next, the vertical lip keypoints (i.e., K19-Y and K20-Y) are scaled by the normalized audio envelope $\mathbf{env} \in [0, 1]$, introducing alignment with speech intensity. Finally, we add Gaussian noise with standard deviation 0.1 only to non-lip dimensions, injecting diversity without corrupting articulation. This initialization produces expressive yet smooth motion that guides the model toward semantically meaningful behavior, balancing stochastic diversity with expressive controllability.

**Multi-Head Loss Balancing.** To supervise disentangled heads, we adopt a unified multi-head flow-matching loss to supervise predicted velocity field $\mathbf{v}_h$, for head $h \in \{\text{emo}, \text{talk}, \text{pose}\}$ conditioned on task-specific priors as defined in Eq. (4) and initial distributions in Eq. (5), defined as:

$$\mathcal{L}_{\text{flow}}(h) = \sum_{h \in \{\text{emo}, \text{lip}, \text{pose}\}} \lambda_h \cdot \mathbb{E}_{\mathbf{z}_0^h, \mathbf{z}_1^h, tt}\left[\left\|\mathbf{v}_h(\mathbf{z}_t^h, tt) - (\mathbf{z}_1^h - \mathbf{z}_0^h)\right\|_2^2\right], \quad \mathbf{z}_t^h = (1 - tt)\mathbf{z}_0^h + tt\,\mathbf{z}_1^h,$$

where $tt \sim \mathcal{U}[0, 1]$, and each coefficient $\lambda_h$ balances the relative importance of the corresponding flow head. Then we aggregate the three flow losses by using dynamic weight averaging (DWA) (Liu et al., 2019) and homoscedastic uncertainty (Kendall et al., 2018) as learnable scaling, depicted as:

$$\mathcal{L}(t) = \sum_{j=1}^{3} \frac{w_j^{\text{DWA}}(t)}{2\,\sigma_j^2} \mathcal{L}_{\text{flow}}(h) + \frac{1}{2}\sum_{j=1}^{j} \log \sigma_j^2, \quad w_j^{\text{DWA}}(t) = \exp(\frac{r_j(t)}{T})\Big/\sum_j \exp(\frac{r_j(t)}{T}) \quad (7)$$

where the loss ratios $r_j(t) = \overline{L}_j(t-1)/(\overline{L}_j(t-2) + \varepsilon)$ define the DWA priors (temperature $T > 0$). We optimize $\log \sigma_j^2$ (the $\frac{1}{2}\log\sigma_j^2$ term regularizes and prevents degeneracy) and treat $w_j^{\text{DWA}}(t)$ as a fixed per-epoch prior (no backprop through it). This facilitates interpretable supervision while dynamically balancing task difficulty and uncertainty, achieving more stable and efficient training.

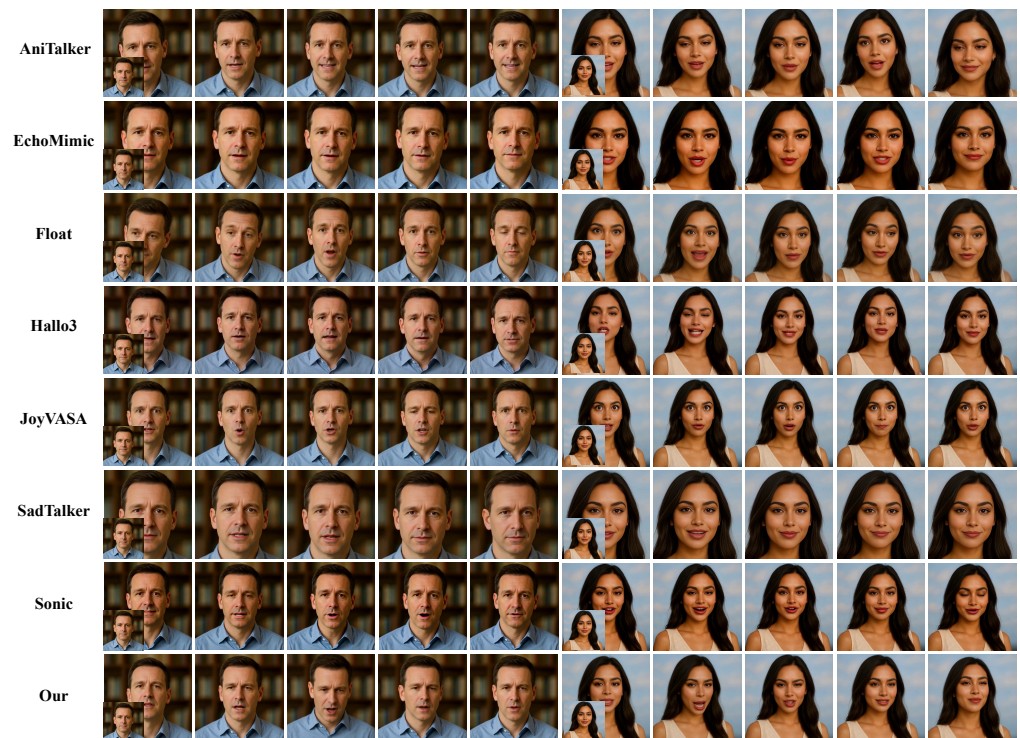

Figure 4: Qualitative comparison with baseline methods. Our XTalker generates expressive portraits with clear facial emotion and head motion while preserving lip-phoneme synchronization.

## 4 EXPERIMENTS

### 4.1 IMPLEMENTATION DETAILS

**Baselines.** We compare our XTalker with seven representative baselines: (1) AniTalker (Liu et al., 2024); (2) EchoMimic (Chen et al., 2025); (3) Float (Ki et al., 2024); (4) Hallo3 (Cui et al., 2024b); (5) JoyVASA (Cao et al., 2024); (6) SadTalker (Zhang et al., 2023); (7) Sonic (Ji et al., 2024).

**Datasets.** We train XTalker on the HDTF dataset (Zhang et al., 2021b), a small-scale but high-quality corpus of 357 talking videos with over 300 identities and 10k sentences. Evaluation is performed on 100 reference images, i.e., 20 synthetic from GPT-5 and 80 real from FFHQ (Karras et al., 2019), and 100 audio clips from CelebV-HQ (Zhu et al., 2022) ranging from 10 to 20 seconds.

**Metrics.** To comprehensively evaluate the performance of talking-head generation, we adopt six metrics (**Appendix A.2**), including Sync-C (Prajwal et al., 2020) and Sync-D for Lip-Sync accuracy, CSIM (Deng et al., 2019) for identity preservation, LPIPS (Zhang et al., 2018) for perceptual similarity, EmoACC (Muru, 2021) for emotion expressiveness accuracy, Head motion variance for diversity and naturalness of head motion, as well as FPS for model inference efficiency.

**Setup.** XTalker is trained for 300k iterations with batch size 1 on a single NVIDIA A100 using AdamW (lr $= 1 \times 10^{-4}$, cosine schedule). We adopt 512 flow-matching and 8 inference steps. The MM-DiT backbone has 8 layers with a hidden size of 512, and each head branch contains 2 layers with the same hidden size. Loss weights are set to 50:1:1 for emotion, talking, and pose, with blending coefficient $\alpha = 0.5$ (Eq. (6)). Evaluation is performed on NVIDIA A100/RTX 4090 GPUs. Additional implementation details are provided in **Appendix A**.

### 4.2 MAIN RESULTS

**Qualitative Evaluation.** We compare our approach with seven baselines, with representative results shown in Figure 4. Notably, our approach consistently produces sharper emotions, accurate

Figure 5: Visualization results of emotion and pose control. These results demonstrate that XTalker can follow user-provided emotion labels and head-rotation curves. More cases in **Appendix G.1**.

Table 1: Comparison of baselines across diverse metrics: lip-phoneme synchronization, emotional diversity, head motion controllability, and inference speed. XTalker achieves real-time performance on an RTX 4090, delivering competitive lip-sync accuracy and superior emotion and motion expressivity. The **best** results are highlighted in bold and the second-best results are underlined.

| | Talking | | | | Emotion | Head Motion | Inference (FPS) | |
| | Sync–C ↑ | Sync–D ↓ | CSIM ↑ | LPIPS ↓ | EmoACC ↑ | Variance ↑ | A100 ↑ | RTX4090 ↑ |
|---|---|---|---|---|---|---|---|---|
| AniTalker | 0.4310 | 9.9988 | 0.8809 | 0.1887 | 0.4467 | 3.3437 | 4.95 | 3.71 |
| EchoMimic | 0.2634 | 10.5873 | 0.8947 | 0.1817 | 0.5559 | 2.1882 | 0.79 | 0.49 |
| Float | **1.0579** | **8.0542** | 0.8255 | 0.2325 | 0.3826 | 9.6554 | 14.17 | OOM‡ |
| Hallo3 | 0.7907 | 10.065 | 0.8420 | 0.0666 | 0.4122 | 0.3425 | 0.13 | OOM‡ |
| JoyVASA | 0.7550 | 9.5100 | 0.8755 | 0.0489 | 0.2142 | 0.4293 | 11.58 | OOM‡ |
| SadTalker | 0.6072 | 9.338 | 0.7881 | 0.2371 | 0.5391 | 5.9567 | 4.97 | 3.06 |
| Sonic | 0.7288 | 8.6079 | 0.9181 | 0.0791 | 0.4146 | 9.6550 | 1.81 | 1.05 |
| Ours | 0.7548 | 8.4644 | **0.9395** | **0.0432** | **0.6476** | **21.2243** | **33.14**† | **28.21** |

†: All speed measurements are taken to complete the whole animation process after model warmup. For our Xtalker, LivePortrait warping one frame requires 21.95 ms, while our MM-DiT network only takes 0.48 ms, about **2.2%** of the warping time. ‡: OOM = Out of Memory.

lip-audio synchronization, and diverse head motion. For instance, in challenging cases with large mouth openings, baseline models (e.g., AniTalker and Float) generate distorted or unstable shapes, whereas our approach maintains clear articulation, showcasing XTalker's capability in improving synchronization and expressiveness. Furthermore, visualizations of emotion and pose control in Figure 5 demonstrate that XTalker can reliably follow user-provided emotion labels and head-rotation curves, confirming its effectiveness in expressive and controllable portrait animation. More visualizations can be found in **Appendix G**.

**Quantitative Evaluation.** We compare our method against baselines across four metrics: talking quality, emotion expressiveness, head motion, and inference speed (Table 1). XTalker achieves competitive lip-sync performance on Sync-C and Sync-D while clearly surpassing others in emotion accuracy, head motion diversity, and efficiency. It obtains the best EmoACC (details in **Appendix D.3**), largest pose variance, and highest FPS (33.14 on A100, 28.21 on RTX4090, over twice the second-best method), while delivering strong identity preservation and visual fidelity (highest CSIM, lowest LPIPS). The slightly lower performance on Sync-C and Sync-D may stem from the limited scale of the training set. These results demonstrate that XTalker achieves real-time performance while providing strong expressivity in portrait animation, underscoring its potential for practical deployment.

## 4.3 ABLATION STUDY

We conduct a quantitative ablation study by decomposing XTalker into different variants, with detailed variant names explained in the table note (Table 2). The results show that the dynamic weighting strategy (DWA and Uncertainty in Eq. (7)) effectively balances multiple loss objectives, mitigating conflicts and achieving a better trade-off among lip-audio synchronization, appearance consistency, emotion retention, and temporal stability. Segment settings $N_{\text{seg}}$ also matter: moving from

Table 2: Ablation study on different heads, training strategies, and initialization settings. Results show that removing modules or using simplified initialization leads to clear performance degradation, while the full multi-head model achieves the best overall results across synchronization, emotion, and pose control. The best results are highlighted in **bold**. EmoACC Details in **Appendix D.3**.

| | Sync–C ↑ | Sync–D ↓ | CSIM ↑ | LPIPS ↓ | Smooth ↑ | EmoACC ↑ | Pose-Variance ↑ |
|---|---|---|---|---|---|---|---|
| All ($\mathcal{H}_t + \mathcal{H}_e + \mathcal{H}_p$) | 0.7548 | **8.4644** | **0.9395** | 0.0432 | **0.9967** | 0.6476 | 21.2243 |
| All w/o DWA | 0.4809 | 8.5912 | 0.7659 | 0.2116 | 0.9954 | 0.6214 | **27.3422** |
| $\mathcal{H}_t$ | 0.6790 | 8.8719 | 0.9310 | **0.0225** | 0.9958 | N/A | 0.4051 |
| $\mathcal{H}_t$ w/o **env** | 0.2716 | 10.2377 | 0.9260 | 0.0265 | 0.9952 | N/A | 0.2038 |
| $\mathcal{H}_t + \mathcal{H}_p$ | **0.8280** | 9.1018 | 0.8920 | 0.0567 | 0.9951 | N/A | 24.2062 |
| $\mathcal{H}_t + \mathcal{H}_e$ | 0.7406 | 8.8440 | 0.9143 | 0.0762 | 0.9949 | **0.7102** | 0.1553 |
| All ($N_{seg} = N$) | 0.6170 | 8.7501 | 0.8031 | 0.2152 | 0.9956 | 0.5014 | 18.5289 |
| All ($N_{seg} = 1$) | 0.4401 | 12.3143 | 0.7572 | 0.2099 | 0.9559 | 0.1830 | 28.5198 |
| All w/o $\text{env}_{init}$ ($N_{seg} = 1$) | 0.3786 | 12.9687 | 0.7408 | 0.2123 | 0.9553 | 0.2068 | 29.7962 |

*All*: our full model. *w/o DWA*: without dynamic weight averaging (Eq. (7)). $\mathcal{H}_t, \mathcal{H}_e, \mathcal{H}_p$: talking, emotion, and pose heads (Eq. (4)). *w/o **env***: removing audio envelope for $\mathcal{H}_t$ head. $N_{seg}$: segment setting (Sect. 3.4), where 1 random noise, 25 **All** model, $N$ first–last frame interpolation. $\text{env}_{init}$: envelope-based initialization for mouth keypoints (Sect. 3.4). Visualization in **Appendix G**.

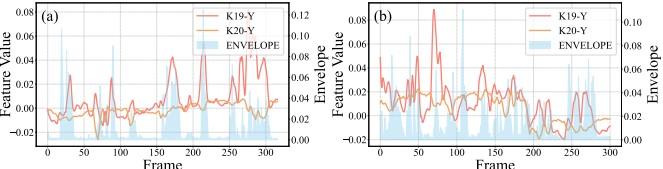
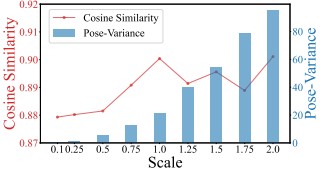

Figure 6: Correlation between lip keypoints and audio envelope in generation videos of XTalker, showing a consistent trend.

Figure 7: Pose controllability under different scaling strengths $\gamma$.

random noise to first-last frame interpolation improves stability, and incorporating audio-envelope $\text{env}_{init}$ guidance further enhances lip synchronization. Moreover, results between $\mathcal{H}_t$, $\mathcal{H}_e$ and $\mathcal{H}_p$ demonstrate that removing the pose and emotion heads yields more stable but less expressive results, whereas adding the pose head enhances motion expressivity at a slight cost to stability. Overall, these findings highlight the importance of multi-head design and dynamic weighting for achieving both stability and expressivity in portrait animation.

### 4.4 ANALYSIS

**Lip Envelope Correlation.** Figure 6 shows the correlation between lip keypoints and the audio envelope in videos generated by XTalker. Although the correlation is weaker than that in real videos (Figure 2), the generated results still follow a consistent trend, indicating that XTalker effectively captures the temporal dynamics of speech to produce reasonable lip–audio synchronization.

**Pose Scaling.** Figure 7 reports the effect of scaling strength $\gamma$ on head motion (visualization in Appendix E.2). As the scaling parameter increases, the variance of head poses grows nearly linearly, while the cosine similarity between generated and driving motions first increases and then stabilizes. These results confirm that integrating the pose head provides a controllable mechanism for adjusting motion amplitude, demonstrating that XTalker enables intuitive and robust pose manipulation.

## 5 CONCLUSION

In this work, we addressed the challenge of achieving expressive audio-driven portrait animation under limited supervision. By revisiting parameter representations, we systematically decomposed facial motion into three interpretable subspaces: lip synchronization, emotional diversity, and head motion controllability, forming the foundation of our proposed framework XTalker. Built upon a unified MM-DiT backbone with three lightweight heads, XTalker provides fine-grained control over facial behaviors while retaining a compact design. Extensive experiments demonstrate that XTalker not only achieves competitive lip–audio synchronization but also delivers superior emotion expressivity and pose controllability, all at real-time speed on a single RTX 4090 GPU.

## ETHICS STATEMENT

Our work on talking portrait animation is a double-edged sword. While it enables positive applications in digital humans, education, and accessibility, it also carries the risk of misuse, such as creating deceptive or harmful content. We emphasize that the technology itself is neutral, and potential harm stems from inappropriate use. To mitigate these risks, we will incorporate safeguards and content review mechanisms when releasing our models and code, ensuring responsible and ethical dissemination.

## REPRODUCIBILITY STATEMENT

We have provided all major model parameters, architectural details, and training procedures necessary to reproduce our results. We plan to release the full codebase (i.e., dataset preprocessing, training, and inference code) and pretrained models under the Apache 2.0 license, but will add appropriate safeguards before open-sourcing to address ethical considerations.

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

# A IMPLEMENTATION DETAILS

## A.1 BASELINES

We compare our XTalker with seven representative baselines:

- **AniTalker** (Liu et al., 2024): employs identity-disentangled motion encoders to generate expressive talking portraits from audio, capturing both facial expressions and head dynamics.
- **EchoMimic** (Chen et al., 2025): enables controllable talking-face generation by conditioning on editable facial landmarks, allowing fine-grained user intervention.
- **Float** (Ki et al., 2024): leverages flow-matching in a latent motion space to synthesize temporally coherent, emotion-aware facial animations driven by speech.
- **Hallo3** (Cui et al., 2024b): adopts a diffusion-transformer framework to produce high-fidelity, temporally consistent image and face animations.
- **JoyVASA** (Cao et al., 2024): disentangles static 3D facial structure from dynamic motion and employs diffusion modeling to generate realistic and controllable talking faces from audio.
- **SadTalker** (Zhang et al., 2023): predicts 3DMM expression and head-pose coefficients from speech and maps them to 3D keypoints for synchronized face animation.
- **Sonic** (Ji et al., 2024): introduces an audio-driven paradigm that exploits global audio perception, disentangling intra- and inter-clip cues to achieve natural, temporally consistent, and diverse talking-face animations without auxiliary visual signals.

## A.2 DETAILED METRICS

We adopt six metrics to comprehensively evaluate talking portrait generation:

Table 3: Spatial behaviors of 21 portrait keypoints along the X-, Y-, and Z-axes.

| Index | Generate Area | X-Axis Adjustment | Y-Axis Adjustment | Z-Axis Adjustment |
|---|---|---|---|---|
| K0 | Deformation: Upper Forehead | Upper Forehead position move left and right | Upper Forehead position move up and down | Upper Forehead generation |
| K1 | Deformation: Forehead | Forehead position generation and affects the frown and raise of the glabella | Forehead position generation and affects the press down and raise of the whole eyebrow | Forehead position generation |
| K2 | Deformation: Eyebrow | Glabella frown and raise at the center | Eyebrow press down and raise as a whole | Eyebrow generation |
| K3 | Deformation: Chin (left) | Chin left cheek move left and right | Chin left cheek move up and down | Chin left cheek generation |
| K4 | Deformation: Sideburn (right) | Right sideburn move left and right | Right sideburn move up and down | Right sideburn generation |
| K5 | Deformation: Neck | Neck drives head move left and right | Neck drives head move up and down | Neck generation |
| K6 | Non-influential area | Changes within the range have no substantial effect | Changes within the range have no substantial effect | Changes within the range have no substantial effect |
| K7 | Deformation: Chin (right cheek) | Chin right cheek move left and right | Chin right cheek move up and down | Chin right cheek generation |
| K8 | Non-influential area | Changes within the range have no substantial effect | Changes within the range have no substantial effect | Changes within the range have no substantial effect |
| K9 | Non-influential area | Changes within the range have no substantial effect | Changes within the range have no substantial effect | Changes within the range have no substantial effect |
| K10 | Deformation: Neck & Sideburn (right) | Right sideburn move left and right | Right sideburn move up and down | Neck becomes thicker / Neck becomes thinner |
| K11 | Action: Left eye | Left eye looks left and right | Left eye looks up and down | Left eye generation |
| K12 | Non-influential area | Changes within the range have no substantial effect | Changes within the range have no substantial effect | Changes within the range have no substantial effect |
| K13 | Action: Left eye | Left eye looks left and right | Left eye looks up and down | Left eye generation |
| K14 | Action: Mouth | Left mouth corner raise / Right mouth corner raise | Smile to O-shape | Mouth generation |
| K15 | Action: Eye (both) | Eye looks left and right | Eye looks up and down | Eye generation |
| K16 | Action: Eye | Outer canthus raise or lower | Inner canthus raise or lower | Eye generation |
| K17 | Action: Mouth & Nose | Left mouth corner raise or press down | Nose becomes wider or narrower | Pouting or grinning |
| K18 | Action: Eye (both) | Eye looks left and right | Eye looks up and down | Eye generation |
| K19 | Action: Mouth(Lower lip) | Lower lip move left and right | Lower lip moves up and down | Lower lip protrude and retract |
| K20 | Action: Mouth(Upper lip) | Upper lip move left and right | Upper lip moves up and down | Upper lip protrude and retract |

- **Sync-D** (Prajwal et al., 2020): quantifies the temporal discrepancy between audio and lip motion within a temporal window, serving as the primary measure of lip–audio alignment and forming the basis for the Sync-C stability metric.
- **Sync-C** (Prajwal et al., 2020): measures the consistency of lip–audio synchronization by computing the difference between the median and mean of Sync-D values within a temporal window, capturing stability of alignment.
- **CSIM** (Deng et al., 2019): computes cosine similarity between identity embeddings of generated and reference images, ensuring identity preservation.
- **LPIPS** (Zhang et al., 2018): evaluates perceptual similarity by comparing deep visual features, capturing human-perceived quality beyond pixel-level metrics.
- **EmoACC** (Muru, 2021): measures expression classification accuracy using our recognition model (Appendix D.2), indicating whether intended emotions are faithfully conveyed.
- **Pose-Variance**: calculates the variance of the face center coordinates across frames, providing an objective measure of head motion diversity and naturalness by penalizing static heads while rewarding realistic movement.

### A.3 MODEL STRUCTURE

To enable fine-grained control over different aspects of portrait animation, XTalker introduces three specialized head branches with detailed structures:

- **Talking Head Branch**: takes hidden states together with audio–envelope features and predicts the motion embedding corresponding to facial expressions (i.e., all latent dimensions except those for head pose). It is implemented as a lightweight 2-layer Transformer with 8 attention heads and a hidden size is 512.

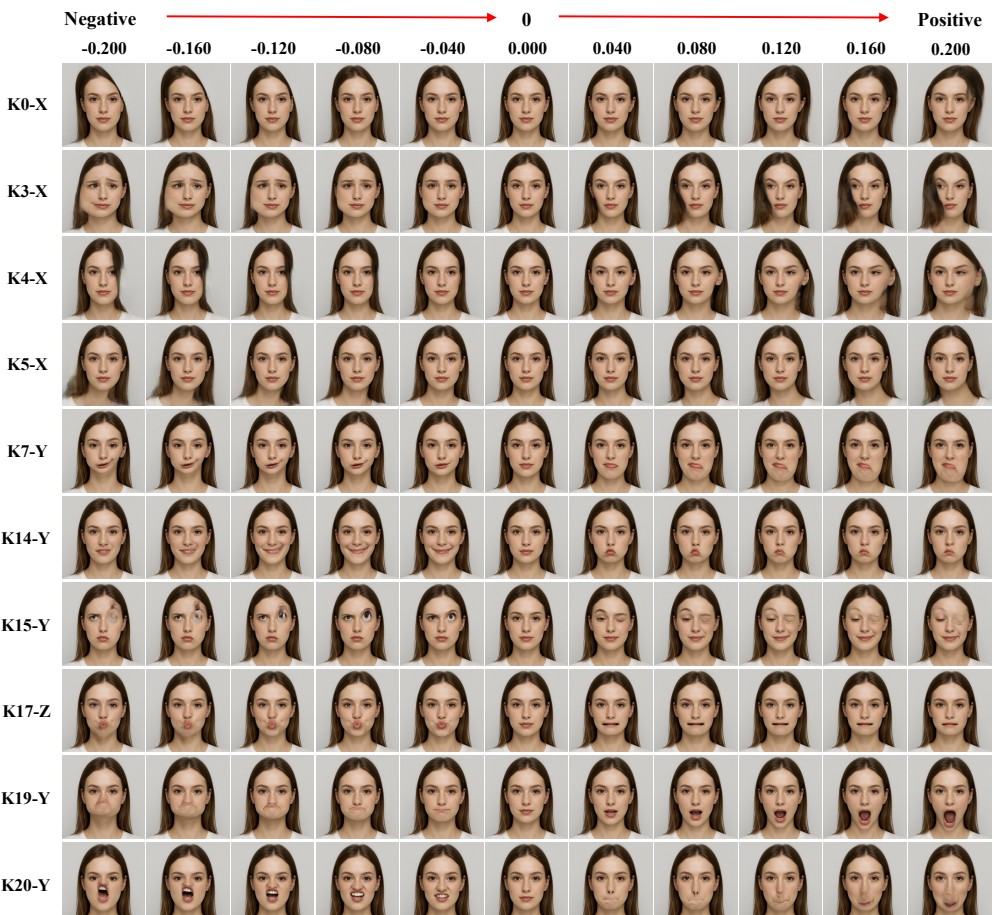

Figure 8: Spatial behaviors of 10 typical face-related keypoints with different values.

- **Emotion Head Branch**: processes averaged hidden states together with reference emotion and one-hot embeddings, and predicts emotion embeddings of dimension 63 through a 2-layer Transformer with 8 attention heads and hidden size 512.
- **Pose Head Branch**: fuses averaged hidden states and curve embeddings to regress 2D pose parameters (pitch, yaw), implemented as a 2-layer Transformer with 8 attention heads and a reduced hidden size of 128.

## B PORTRAIT KEYPOINTS ANALYSIS

In this section, we conduct a detailed analysis of 21 portrait keypoints to assess their contributions to portrait motion along the X-, Y-, and Z-axes, with results summarized in Table 3 and Figure 8. Keypoints around the mouth (e.g., K14, K17, K19 and K20) are found to be critical for lip synchronization and speech articulation, as their local displacements directly shape phoneme-dependent lip movements. Eye-related keypoints (K11, K13, K15, K16 and K18) are also essential, capturing gaze shifts and blinking patterns that enhance naturalness and engagement. By contrast, regions such as the neck and sideburns mainly modulate head pose dynamics, supporting controllable head motion but contributing little to speech-driven synchronization. Several indices (e.g., K6, K8, K9 and K12) show negligible influence, suggesting redundancy that can be exploited for disentanglement and model efficiency. Overall, this spatial analysis highlights the decisive roles of mouth and eye regions in realistic talking-head generation, and motivates our noise initialization strategy for fine-grained control.

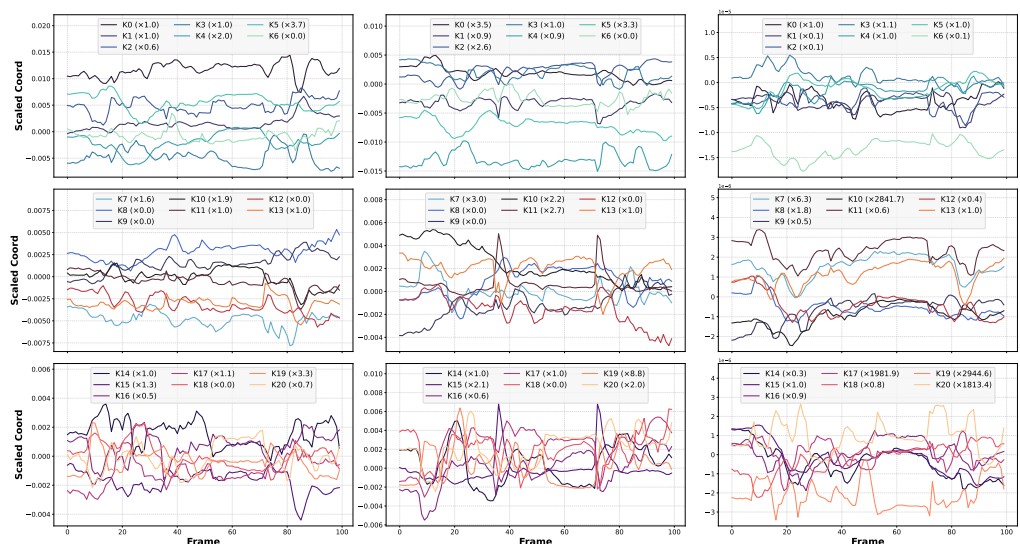

Figure 9: Temporal patterns of 21 portrait keypoints across time frames.

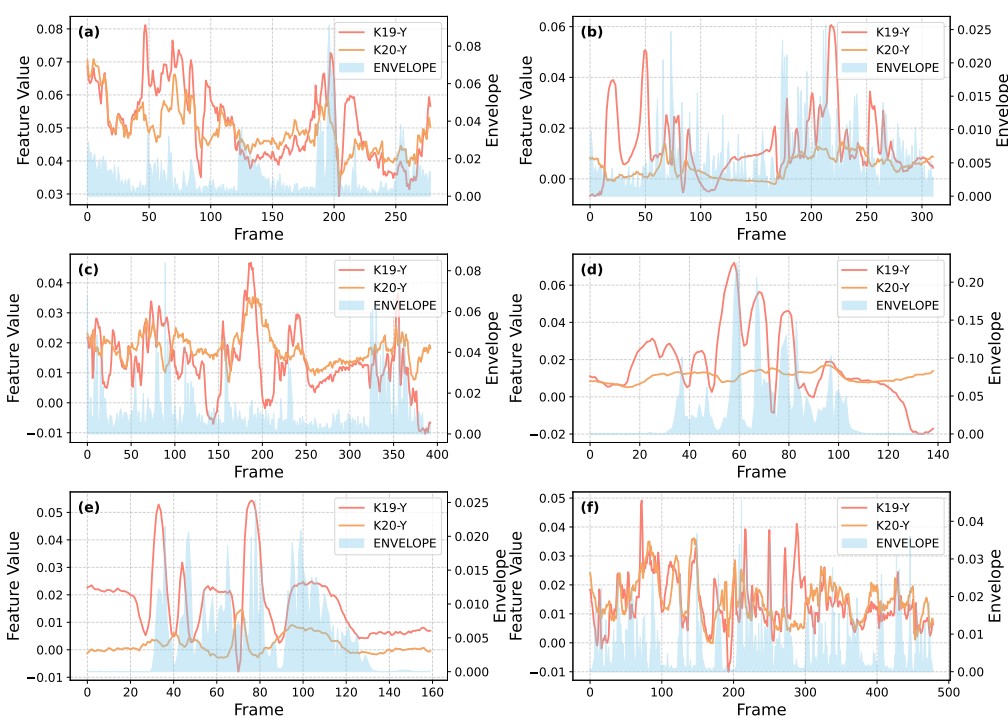

Figure 10: Correlation between lip and audio envelope in **raw video samples**. The mouth keypoints exhibit high correlation with the audio envelope. K19-Y denotes the Y-axis of keypoint 19, and K20-Y denotes the Y-axis of keypoint 20. Notably, these two keypoints correspond to the upper and lower lips, which move in opposite directions during audio articulation.

## C  VIDEO KEYPOINTS ANALYSIS

### C.1  KEYPOINTS ANALYSIS

To further analyze the dynamic behavior of portrait keypoints, we visualize their trajectories across frames in Figure 9. The plots reveal that mouth-related keypoints exhibit the largest variations,

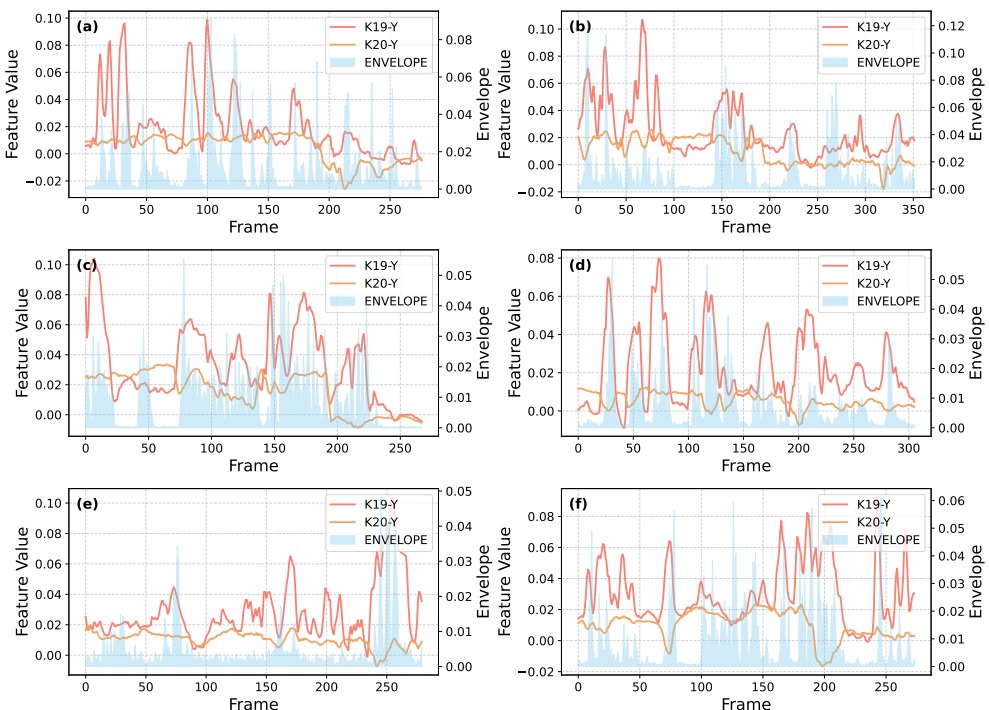

Figure 11: Correlation between lip and audio envelope in **XTalker generated samples**. K19-Y denotes the Y-axis of keypoint 19, and K20-Y denotes the Y-axis of keypoint 20. Similar to Figure 10, the generated trajectories still follow the overall envelope trend, indicating that our method effectively learns and preserves the articulation patterns aligned with speech dynamics.

closely following the speech envelope and reflecting phoneme-dependent articulation. Eye keypoints display moderate but distinct oscillations corresponding to blinking and gaze shifts, which are indispensable for natural expressiveness. In contrast, non-influential keypoints maintain relatively flat trajectories, reinforcing their limited contribution to controllable portrait motion. Notably, the analysis also indicates pronounced long-term fluctuations together with short-term high-frequency jitters, consistent with our training strategy of applying random linear interpolation every 25 frames combined with local jitter. Overall, this temporal analysis highlights that both local sensitivity and multi-scale variability of keypoints are crucial for achieving realistic and high-quality talking-head generation.

### C.2 MOUTH KEYPOINTS FROM RAW VIDEOS

Since lip movements are the most critical factor in determining the perceived quality of talking head generation, we further examine the correlation between mouth-related keypoints (K19 and K20) and the envelope of the driving audio. As shown in Figure 10, the temporal variations of these two keypoints exhibit strong alignment with the speech envelope across multiple video samples. This indicates that local mouth dynamics are tightly synchronized with the prosodic patterns of speech, such as energy fluctuations and phoneme transitions. Such high correlation not only validates the importance of mouth keypoints in driving lip synchronization but also highlights their role as a reliable proxy for modeling speech-articulatory coupling in talking head generation, and thus inspires our envelope initialization strategy.

### C.3 MOUTH KEYPOINTS FROM GENERATED VIDEOS

In addition to real recordings, we also extract lip keypoints from videos generated by XTalker for correlation analysis. Figure 11 illustrates the correlation between these keypoints and the audio envelope. Although weaker than that in real videos, the generated results still exhibit a consistent

Table 4: MSE performance of Emotion Expression Transformer across different emotions.

| Acc ↑ | Angry ↑ | Disgusted ↑ | Afraid ↑ | Happy ↑ | Neutral ↑ | Sad ↑ | Surprised ↑ |
|-------|---------|-------------|----------|---------|-----------|-------|-------------|
| 1.26e-06 | 1.16e-06 | 1.35e-06 | 1.24e-06 | 1.22e-06 | 1.19e-06 | 1.29e-06 | 1.36e-06 |

Table 5: Emotion recognition accuracy by the Emotion Recognition model across the KDEF and HDTF datasets.

| Acc ↑ | Angry ↑ | Disgusted ↑ | Afraid ↑ | Happy ↑ | Neutral ↑ | Sad ↑ | Surprised ↑ |
|-------|---------|-------------|----------|---------|-----------|-------|-------------|
| 0.9150 | 0.8427 | 0.8578 | 0.9587 | 0.9989 | 0.9978 | 0.7952 | 0.9531 |

trend, indicating that XTalker can effectively capture the temporal dynamics of speech to produce reasonable lip–audio synchronization. This discrepancy may stem from the lack of fine-grained phoneme-level supervision, which can smooth rapid transitions, as well as from the stochastic nature of generative processes that tend to average out subtle variations. Nevertheless, the alignment with the audio envelope confirms that XTalker has learned a robust mapping from acoustic cues to articulatory motion, ensuring that synthesized talking heads remain both intelligible and natural across diverse speech inputs.

## D EMOTION EXPRESSION

### D.1 EMOTION EXPRESSION TRANSFORMER

To support expression generation within the overall pipeline, we design a lightweight DiT-style Transformer regressor that maps source facial keypoints and target emotion labels to target keypoint embeddings. The source vector and one-hot emotion label are first concatenated, linearly projected to a 128-dimensional latent space, and enriched with 1D sinusoidal positional encoding. This representation is processed by a 2-layer Transformer encoder with 8 attention heads, then mapped back to the keypoint dimension via a linear layer. For training, we construct paired expressions per subject from the KDEF dataset (Lundqvist et al., 1998), optimizing with mean squared error using Adam ($1 \times 10^{-4}$ learning rate, batch size 32, 200 epochs). In inference, the model generates target facial keypoints, which are evaluated against ground truth to compute reconstruction error. Experiments show consistently low MSE across expressions and qualitatively accurate generation in Table 4. We further synthesize diverse expressions on the HDTF dataset (Zhang et al., 2021b), providing XTalker with emotion-conditioned keypoint embeddings that combine original landmarks with target emotion labels, which are subsequently used in the flow-matching training pipeline. Although performance may degrade under poor illumination or ambiguous expressions due to the limited scale of KDEF, the trained model remains sufficiently robust to serve as a prediction tool within our framework.

### D.2 EMOTION EXPRESSION RECOGNITION

For emotion recognition, we adopt an InceptionV3-based model (InceptionV3FER) to evaluate the expressive quality of generated talking portraits. The backbone is initialized with ImageNet pre-trained weights and adapted for seven-class recognition (Afraid, Angry, Disgusted, Happy, Neutral, Sad, Surprised) by replacing the fully connected layers with a sequence of linear layers to transform classes (1024→512→256→128→7). During training, backbone parameters are frozen for the first five epochs and progressively unfrozen thereafter, optimized with cross-entropy loss (including a weighted auxiliary output, weight 0.4) using Adam and a layer-wise learning rate schedule. Training is conducted on the KDEF and HDTF datasets (Use the emotion expression transformer to build images with different emotions), with images resized to 326×326, batch size 32, and an 80/20 train-validation split. The model achieves high accuracy across all seven emotions (shown in Table 5), with quantitative metrics consistent with qualitative observations, confirming its reliability in evaluating the expressivity of generated samples. We adopt this model as the recognition to calculate **the metric "EmoACC" in Sect. 4.1**.

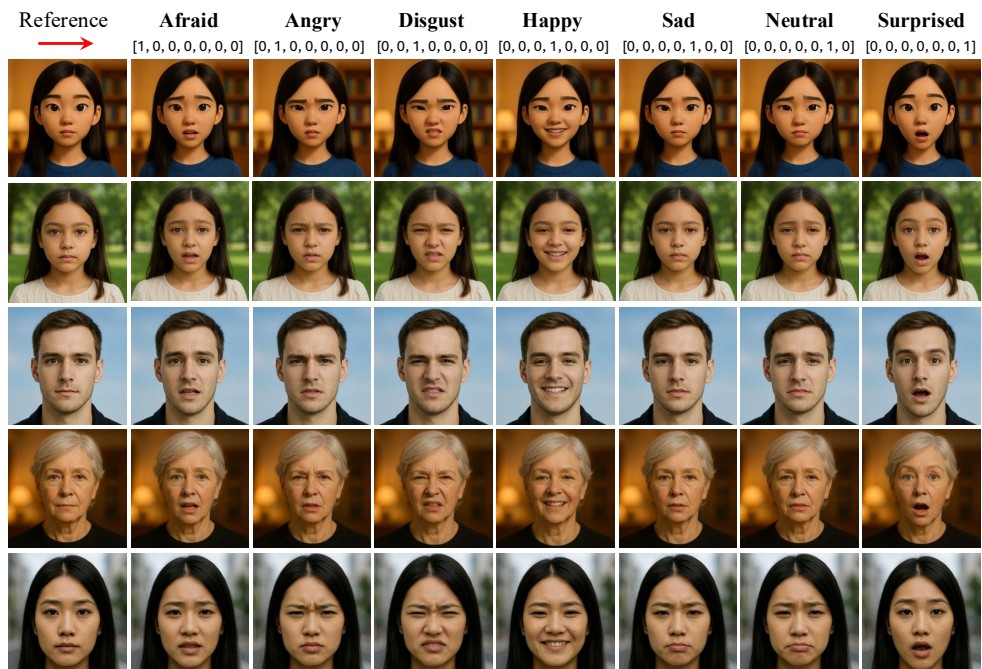

| Reference | Afraid | Angry | Disgust | Happy | Sad | Neutral | Surprised |
| → | [1, 0, 0, 0, 0, 0, 0] | [0, 1, 0, 0, 0, 0, 0] | [0, 0, 1, 0, 0, 0, 0] | [0, 0, 0, 1, 0, 0, 0] | [0, 0, 0, 0, 1, 0, 0] | [0, 0, 0, 0, 0, 1, 0] | [0, 0, 0, 0, 0, 0, 1] |

Figure 12: XTalker converts neutral reference images into diverse emotional portraits conditioned on different labels, demonstrating its ability to generate expressive and controllable facial variations.

Table 6: EmoACC by emotion recognition model across different baselines, corresponding to Table 1. The **best** results are highlighted in bold and the second-best results are underlined.

| Method | Acc ↑ | Angry ↑ | Disgusted ↑ | Afraid ↑ | Happy ↑ | Neutral ↑ | Sad ↑ | Surprised ↑ |
|---|---|---|---|---|---|---|---|---|
| AniTalker | 0.4467 | 0.2034 | 0.3136 | **0.7797** | 0.7034 | 0.3475 | 0.1102 | 0.6695 |
| EchoMimic | 0.5559 | 0.4521 | 0.4966 | 0.6747 | 0.5500 | 0.6857 | 0.4643 | 0.5679 |
| Float | 0.3826 | 0.3559 | 0.4633 | 0.2825 | 0.3729 | 0.6045 | 0.5198 | 0.0791 |
| Hallo3 | 0.3425 | 0.3855 | 0.2771 | 0.1446 | 0.1687 | **0.8795** | 0.4819 | 0.0602 |
| JoyVASA | 0.2142 | 0.0449 | 0.0473 | 0.4886 | 0.0256 | 0.2364 | 0.0615 | 0.5957 |
| SadTalker | 0.5391 | 0.4972 | 0.5367 | 0.2994 | 0.8249 | 0.8249 | 0.2825 | 0.5085 |
| sonic | 0.4146 | 0.0312 | 0.2500 | 0.5500 | 0.6875 | 0.5938 | 0.2812 | 0.7692 |
| Ours | **0.7597** | **0.7321** | **0.7757** | 0.5514 | **0.9065** | 0.7009 | **0.8318** | **0.8193** |

### D.3 EMOTION EXPRESSION VISUALIZATIONS

To evaluate the emotion expression capability of XTalker, we first visualize its ability to convert neutral reference portraits into diverse emotional images. As shown in Figure 12, XTalker successfully synthesizes a wide range of expressive faces conditioned on different emotion labels, demonstrating controllable and realistic expression transfer while preserving identity consistency.

Beyond qualitative results, we quantitatively assess expressivity using our emotion recognition model. Table 6 reports the emotion classification accuracy (EmoACC) across different baselines. Compared with existing approaches, XTalker achieves the highest overall accuracy (0.7597) and consistently outperforms others across most emotion categories, particularly in generating *Happy*, *Sad*, and *Surprised* expressions. These results confirm that XTalker not only produces visually convincing emotional variations but also enhances discriminability across multiple emotions, validating its effectiveness in expression-controllable talking portrait generation.

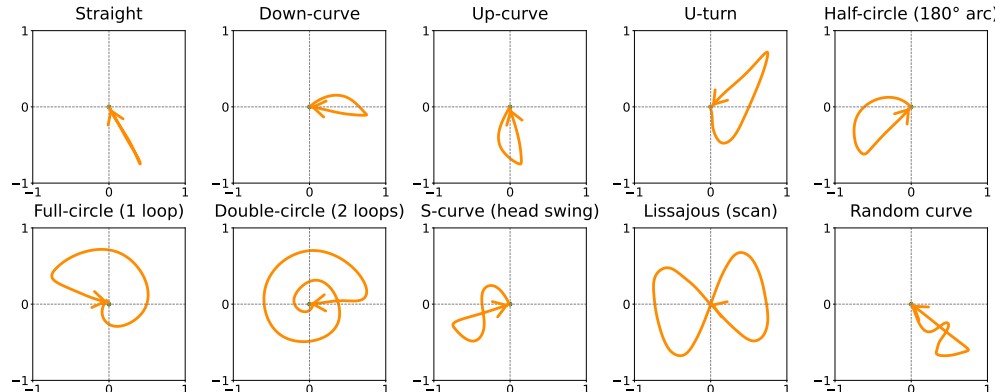

Figure 13: Several examples of curves generated by ChatGPT-5. To ensure temporal continuity in audio-driven portrait animation, where videos are composed of concatenated segments, each curve is designed to gradually return to the origin in the final 20% of its trajectory.

## E  LLM-GUIDED CURVE-POSE SYNTHESIS

### E.1  LLM-GUIDED CURVE-POSE PREPARATION

To enable controllable head motion, we design a curve-to-pose module that translates user-drawn or simulated curves into Pitch–Yaw trajectories, while assigning random or zero values to Roll to preserve natural variation. Specifically, each input curve is represented as a set of 2D coordinates centered at the image origin and then mapped to pose increments. A lightweight *pose head* integrates these curve embeddings with backbone features through a 2-layer DiT block (8 attention heads, hidden size $H/4$), producing frame-wise pose predictions $\mathbf{v}_p \in \mathbb{R}^{N \times 2}$. An additional scale parameter allows us to adjust overall motion intensity, ensuring flexibility across different use cases.

To construct large-scale training and evaluation data, we leverage a language model (e.g., GPT-5 API) to generate diverse curve specifications and corresponding 2D point sets from natural language prompts. Examples include "a half circle with 100 equidistant points" or "an S-shaped curve with alternating curvature". These curves are normalized to the target coordinate system and designed so that in the final 20% of each trajectory, the head gradually returns to the image center, ensuring natural termination of motion.

Our curve set covers 10 categories with 100 instances each: straight, left turn, right turn, U-turn, half circle, full circle, double circle, S-curve, Lissajous, and random curves. This collection captures both simple and complex motion patterns, from basic sweeps to oscillatory gaze-like movements. The qualitative results of curve-driven head motions are presented in Figure 13, illustrating that our module can effectively transform abstract curve priors into realistic, controllable pose dynamics.

### E.2  POSE MOTION VISUALIZATIONS

Figure 14 demonstrates how XTalker leverages these curve priors in practice: the same reference video is converted into multiple head motion variants under different scaling strengths $\gamma$. Increasing $\gamma$ amplifies the motion intensity, while smaller values produce subtler movements, together highlighting the controllability of our framework. These results complement the cosine-based motion variation analysis in Figure 7, and collectively confirm that XTalker can generate diverse and realistic portrait animations with fine-grained control over head dynamics.

## F  USE OF LARGE LANGUAGE MODELS

Large language models (LLMs) were employed in two complementary aspects of this work. First, in the methodology, we leveraged prompt-based interaction with LLMs (e.g., GPT-5 API) to automatically generate diverse curve specifications, as detailed in Appendix E.1.

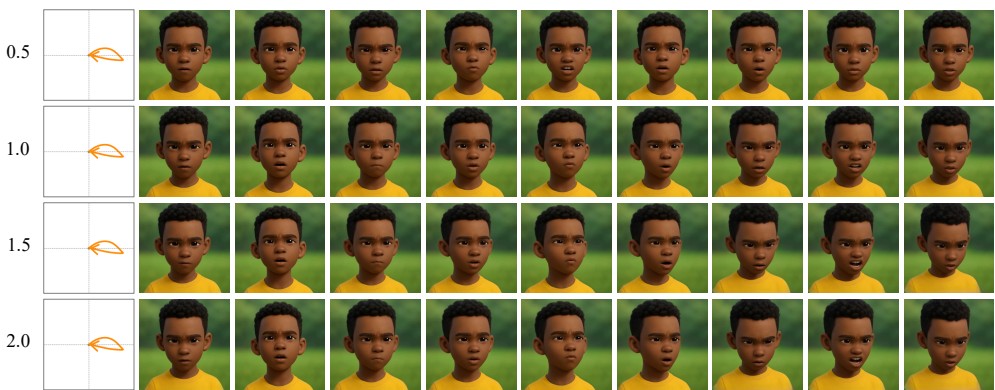

Figure 14: XTalker converts the same video into diverse head motion conditioned on different scaling strength $\gamma$ on head motion, demonstrating its ability to generate portrait animation with motion diversity. This is the further visualizations in Figure 7.

Second, in the preparation of this manuscript, we also utilized LLMs as language assistants to polish the writing. Prompts were provided to improve clarity, conciseness, and readability, while all substantive ideas, experiments, and interpretations remain entirely our own.

## G VISUALIZATION RESULTS

### G.1 XTALKER VISUALIZATIONS

Figure 15 illustrates the ability of XTalker to simultaneously follow user-specified emotion labels and head-rotation curves for expressive and controllable portrait animation. The first column shows the reference images, which remain consistent across different conditions. The second column indicates the target emotion label (e.g., happy, sad, surprised), while the third column visualizes the predefined head-rotation trajectories in pitch–yaw–roll space. The remaining columns present generated video frames under the given emotion–pose conditions. As shown, XTalker accurately captures the requested emotions while maintaining stable lip synchronization and identity consistency. At the same time, the generated head motions precisely follow the provided curve trajectories, even in challenging cases such as large rotations or complex motion patterns. These results highlight the robustness and controllability of the model in jointly manipulating both emotional expression and head dynamics.

### G.2 ABLATION VISUALIZATIONS

Results demonstrate that removing modules or using simplified segment initialization degrades visual quality, with noticeable drops in lip–audio synchronization, emotional expressiveness, or motion stability. In contrast, the full multi-head model with dynamic weighting and envelope guidance achieves the best overall balance across synchronization, emotion retention, and head pose control.

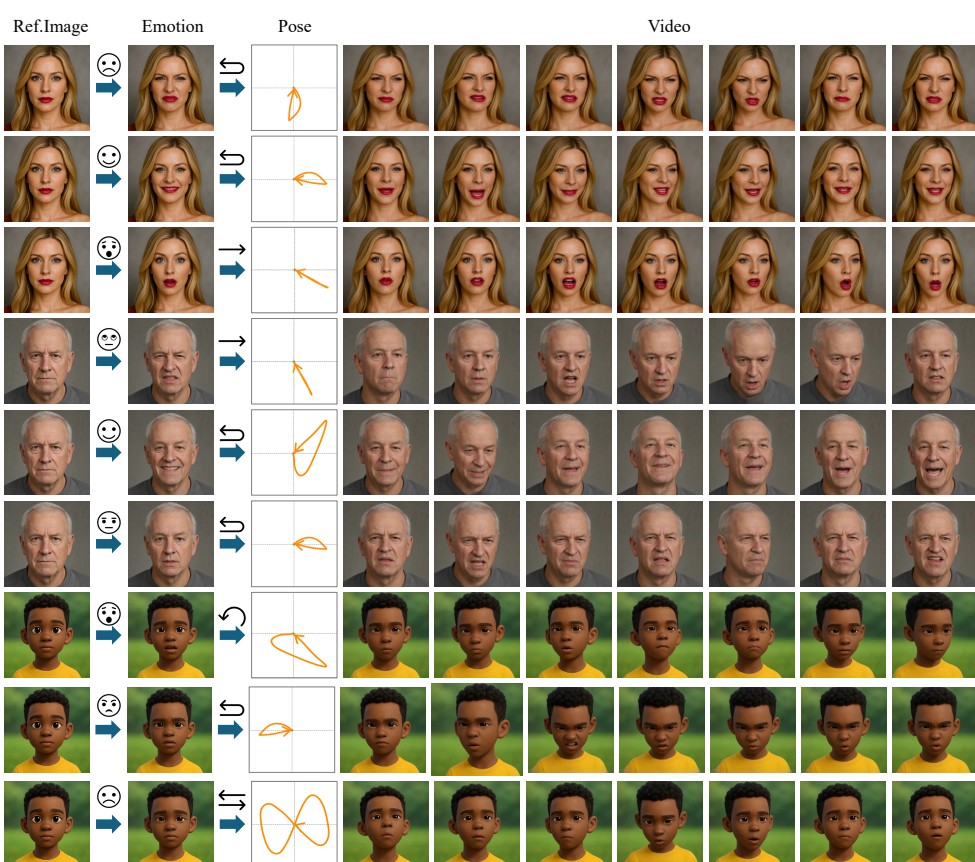

Figure 15: Visualization results of emotion and pose control. These results demonstrate that XTalker can follow user-provided emotion labels and head-rotation curves. These results are the supplement of Figure 5.

ALL
($Ht$ + $He$ + $Hp$)

All w/o DWA

$Ht$

$Ht$ w/o **env**

$Ht$ +$Hp$

$Ht$ +$He$

All ($N_{seg}$ = N)

All ($N_{seg}$ = 1)

All w/o
**env**$_{init}$ ($N_{seg}$ = 1)

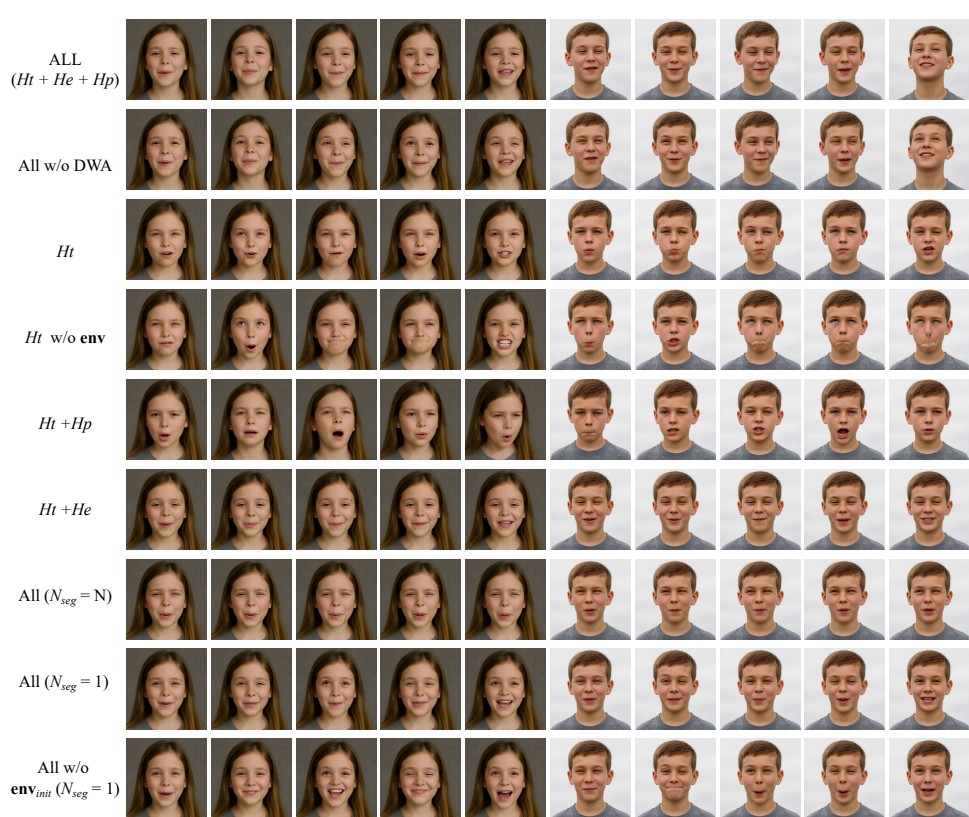

Figure 16: Qualitative results of ablation study. The observed differences across settings are consistent with our expectations, confirming that removing or altering specific components leads to degraded expressivity and controllability compared to the full XTalker framework.

