# OpenReview forum: "XTalker: Turn, Smile, and Speak in Controllable Talking Portrait Animation"
_ICLR.cc/2026/Conference — Submitted to ICLR 2026_

### Official Review · Reviewer_7ymS · 2025-10-23

**Soundness:** 1
**Presentation:** 3
**Contribution:** 2
**Rating:** 2
**Confidence:** 4

**Summary:**

This paper presents XTalker, a framework for audio-driven talking portrait animation. It focuses on disentangling facial dynamics into interpretable subspaces using parameter representations. The method decomposes motion into three components: lip-phoneme synchronization, emotional expression, and head motion. Lip movements are driven by the audio envelope's temporal dynamics. Emotion is modulated through semantic features and labels. Head motion is controlled by user-defined curves. XTalker uses a unified MM-DiT backbone with three lightweight prediction heads for each component. It is designed for fine-grained control and expressivity in low-data settings. The model runs in real-time and achieves competitive lip-sync accuracy. Experimental results show improved expressivity in emotion and motion compared to prior methods.

**Strengths:**

* Disentangled Control: The framework explicitly decomposes facial animation into three interpretable components (lip synchronization, emotion, and head pose) using dedicated lightweight heads, enabling separate control over each aspect.
* Improved Expressivity: Experimental results show that XTalker generates more diverse and expressive head motions and facial emotions compared to several baseline methods, as evidenced by higher variance and emotion accuracy scores.
* Real-time Performance: The model achieves real-time inference speeds (28.21 FPS on an RTX 4090), making it suitable for practical applications requiring low latency.

**Weaknesses:**

* Insufficient experimental results: As a video generation task, this work does not provide generated videos to demonstrate the performance of the proposed method, making it difficult for reviewers to comprehensively assess its performance. Moreover, in the presented qualitative comparison images, all samples have relatively simple backgrounds (either solid-colored or blurred), which fails to adequately reflect the model's overall capabilities. Regarding qualitative evaluation, the method achieves only moderate performance on lip-sync metrics, ranking fourth on Sync–C and second on Sync–D, suggesting potential shortcomings in audio-driven synchronization. In terms of baseline comparisons, this work lacks comparison with more recent approaches such as Ditto[1], FantasyTalking[2] and MultiTalk[3]. Additionally, for emotion control, it does not compare with methods capable of explicit emotion control, such as EAT[4], EDTalk[5], DreamTalk[6], and DICE-Talk[7], making it difficult to evaluate the true performance of this work in emotion control.
* Limited novelty: Audio-driven methods based on parameter representations have been extensively studied; representative prior works include SadTalker[8] and VASA-1[9]. This work builds upon LivePortrait[10] by introducing a DiT network to condition the motion features extracted by LivePortrait on audio, emotion, and pose. This overall approach is highly similar to prior works such as VASA-1, JoyVASA[11], Playmate[12], and Takin-ADA[13], all of which are based on an image animation method (e.g., MegaPortraits[14] for VASA-1, LivePortrait for JoyVASA and Playmate, FaceVid2Vid[15] for Takin-ADA) and employ a DiT network to incorporate various conditioning signals into the features extracted by the animation model. Furthermore, the way this work incorporates emotion conditioning—via a one-hot emotion vector injected through fully connected layers or attention mechanisms—is very similar to that of Playmate and DICE-Talk. Overall, the proposed method does not significantly differ from prior approaches, and its contribution to the community appears limited.
* Missing technical details: The paper does not clearly specify which components of the framework are trainable and which are adopted from existing methods. This lack of clarity makes it difficult to understand the training procedure and implementation details, posing challenges for reproducibility and comprehension.

[1]Li, Tianqi, et al. "Ditto: Motion-space diffusion for controllable realtime talking head synthesis." arXiv preprint arXiv:2411.19509 (2024).

[2]Wang, Mengchao, et al. "Fantasytalking: Realistic talking portrait generation via coherent motion synthesis." arXiv preprint arXiv:2504.04842 (2025).

[3]Kong, Zhe, et al. "Let Them Talk: Audio-Driven Multi-Person Conversational Video Generation." arXiv preprint arXiv:2505.22647 (2025).

[4]Gan, Yuan, et al. "Efficient emotional adaptation for audio-driven talking-head generation." Proceedings of the IEEE/CVF International Conference on Computer Vision. 2023.

[5]Tan, Shuai, et al. "Edtalk: Efficient disentanglement for emotional talking head synthesis." European Conference on Computer Vision. Cham: Springer Nature Switzerland, 2024.

[6]Ma, Yifeng, et al. "DreamTalk: When Emotional Talking Head Generation Meets Diffusion Probabilistic Models." arXiv preprint arXiv:2312.09767 (2023).

[7]Tan, Weipeng, et al. "Disentangle Identity, Cooperate Emotion: Correlation-Aware Emotional Talking Portrait Generation." arXiv preprint arXiv:2504.18087 (2025).

[8]Zhang, Wenxuan, et al. "Sadtalker: Learning realistic 3d motion coefficients for stylized audio-driven single image talking face animation." Proceedings of the IEEE/CVF conference on computer vision and pattern recognition. 2023.

[9]Xu, Sicheng, et al. "Vasa-1: Lifelike audio-driven talking faces generated in real time." Advances in Neural Information Processing Systems 37 (2024): 660-684.

[10]Guo, Jianzhu, et al. "Liveportrait: Efficient portrait animation with stitching and retargeting control." arXiv preprint arXiv:2407.03168 (2024).

[11]Cao, Xuyang, et al. "JoyVASA: portrait and animal image animation with diffusion-based audio-driven facial dynamics and head motion generation." arXiv preprint arXiv:2411.09209 (2024).

[12]Ma, Xingpei, et al. "Playmate: Flexible Control of Portrait Animation via 3D-Implicit Space Guided Diffusion." Forty-second International Conference on Machine Learning.

[13]Lin, Bin, et al. "Takin-ADA: Emotion Controllable Audio-Driven Animation with Canonical and Landmark Loss Optimization." arXiv preprint arXiv:2410.14283 (2024).

[14]Drobyshev, Nikita, et al. "Megaportraits: One-shot megapixel neural head avatars." Proceedings of the 30th ACM International Conference on Multimedia. 2022.

[15]Wang, Ting-Chun, Arun Mallya, and Ming-Yu Liu. "One-shot free-view neural talking-head synthesis for video conferencing." Proceedings of the IEEE/CVF conference on computer vision and pattern recognition. 2021.

**Questions:**

* Could the authors provide additional implementation details, such as the training pipeline, which modules are trained end-to-end versus those fixed or adopted from pre-trained models, and the step-by-step training procedure?
* Would it be possible to provide video inference results via an anonymous url? This would help reviewers better assess the actual performance of the model.
* In certain cases, LivePortrait[1] tends to produce artifacts in the mouth region, particularly when the subject is pursing their lips. Several existing methods that build upon LivePortrait for audio-driven animation exhibit similar issues. Does the proposed method encounter the same problem? If so, could the authors provide a discussion on this limitation?

[1]Guo, Jianzhu, et al. "Liveportrait: Efficient portrait animation with stitching and retargeting control." arXiv preprint arXiv:2407.03168 (2024).

---

> ### Author Response · Authors · 2025-11-28
>
> Dear Reviewer.
>
> We appreciate the reviewers’ efforts in evaluating our submission.
>
> We have updated the work by adding qualitative comparison videos that better demonstrate the method’s performance. We hope the new evidence will support a more complete academic assessment and help clarify earlier concerns.
>
> We will continue revising the paper following the suggestions provided.
>
> Best regards,
>
> Authors

---

### Official Review · Reviewer_akiD · 2025-10-27

**Soundness:** 2
**Presentation:** 2
**Contribution:** 1
**Rating:** 2
**Confidence:** 4

**Summary:**

The paper proposes XTalker, a fast and controllable audio-driven portrait animation framework that generates talking faces from a single image and audio. It decomposes facial motion into three interpretable subspaces—lip synchronization, emotion modulation, and head pose control—and employs a unified MM-DiT backbone with three lightweight heads for each component.

**Strengths:**

1. Decomposes facial dynamics into three components: lip synchronization, emotion, and head motion.
2. Designs a lightweight framework with three dedicated branches to synthesize lip movements, emotional expressions, and head motion.

**Weaknesses:**

1. Missing supplementary video. This is a talking-head generation method, yet no supplementary video is provided. Without videos, it is difficult to assess the temporal consistency and synchronization quality of the generated results.
2. Unclear disentanglement supervision. The authors claim that expression, emotion, and head pose are disentangled, but no explicit supervision (e.g., cycle consistency loss) is applied. Moreover, the final results fuse the outputs of three heads together, making it unclear whether these factors remain disentangled in practice.
3. Limited improvement over existing work. The LivePortrait implementation already supports independent control of lip, eye, and head movements. The proposed method appears to offer only marginal improvements over LivePortrait.
4. Potential metric error. In Table 1, the reported synchronization score computed with SyncNet ranges between 0–1, whereas most existing works report Sync-C scores typically within 3–8. This discrepancy raises concerns about possible inconsistencies in metric computation or reporting.

**Questions:**

None

**Details Of Ethics Concerns:**

This technology could be misused to create realistic talking-face deepfakes without the consent of the individuals involved. Such misuse poses potential risks to privacy, personal reputation, and public trust in digital media. Given the increasing accessibility of generative models, the authors should discuss safeguards or watermarking mechanisms to prevent malicious applications.

---

> ### Author Response · Authors · 2025-11-28
>
> Dear Reviewer.
>
> We appreciate the reviewers’ efforts in evaluating our submission.
>
> We have updated the work by adding qualitative comparison videos that better demonstrate the method’s performance. We hope the new evidence will support a more complete academic assessment and help clarify earlier concerns.
>
> We will continue revising the paper following the suggestions provided.
>
> Best regards,
>
> Authors

---

### Official Review · Reviewer_mAg6 · 2025-10-31

**Soundness:** 2
**Presentation:** 2
**Contribution:** 2
**Rating:** 6
**Confidence:** 4

**Summary:**

XTalker is a method for turning a single portrait photo into a talking video where you can control how the face moves and expresses emotion. Given an input image and an audio clip, the system makes the person in the photo speak in sync with the audio, and it allows the user to specify things like emotion or a head turn.

**Strengths:**

- The approach explicitly breaks down facial animation into three interpretable components: lip synchronization, emotional expression, and head motion. By isolating these subspaces, the system can finetune lip movements, expressions, and head pose separately.
- XTalker’s architecture is efficiently designed with a unified diffusion transformer backbone (MM-DiT) and three control heads (one each for lips, emotion, and pose). This design means the model encodes the audio and image together, then applies specialized small networks for each type of motion.
- The system runs in real time (tested on a single RTX 4090 GPU) while still producing expressive results.The paper reports that XTalker achieves competitive lip–audio synchronization and delivers more varied emotions and head movements than previous methods.

**Weaknesses:**

- While XTalker’s lip-sync is good, it is not the SOTA on every metric. In the quantitative comparison (Table 1), its lip synchronization scores are behind the top baseline on some measures.
- The framework assumes the user will specify an emotion label and a head motion trajectory curve for each animation: it does not automatically infer emotion or head movements from the audio (eg. Speech-Driven Emotional Disentanglement for 3D Face Animation ICCV2023, Emotional Speech-driven 3D Body Animation via Disentangled Latent Diffusion CVPR2024). It could be seen as a limitation, if no emotion/pose input is given, the system wouldn’t add those aspects by itself.
- The multi-task nature of XTalker (synchronizing lips while also handling expression and motion) makes training requiring careful loss balancing. The authors employ a dynamic weighting strategy to balance the different objectives. This indicates that the system required significant tuning to prevent one aspect (such as head motion) from degrading another (like lip accuracy or video stability). Such complexity might be harder to reproduce.

**Questions:**

- The paper uses a Large Language Model to generate predefined head motion curves. What kind of prompts or data were given to this LLM, and how do we know the resulting motion curves look natural?
- The training pipeline includes an Emotion Expression Transformer that creates target facial keypoint embeddings for a desired emotion. How confident can we be that these generated “happy” or “angry” expressions truly correspond to those emotions? For example, did the authors perform any validation (maybe via an emotion classifier or human judgment) to ensure the transformer’s outputs match genuine expressions?
- XTalker relies on user-provided head motion curves to animate head turns. What happens if a user does not supply a custom head trajectory? Does the system default to no head movement, or could it infer a reasonable head motion from the audio or learned patterns? Some previous works generate subtle head movements from audio alone for realism. It would be interesting to know if the authors considered an automatic mode for head motion (and emotion) when explicit control inputs are not given.

---

> ### Author Response · Authors · 2025-11-28
>
> Dear reviewer,
>
> We would like to express our genuine appreciation for the encouraging review and recognition of the work.
>
> As the only positive evaluation received so far, your comments were particularly meaningful for us, and we thank you for acknowledging the contribution and potential of our approach.
>
> We will follow your suggestions to polish our draft for the next conference.
>
> Best wishes,
>
> Authors.

---

### Official Review · Reviewer_sJRq · 2025-11-01

**Soundness:** 2
**Presentation:** 2
**Contribution:** 2
**Rating:** 0
**Confidence:** 4

**Summary:**

The paper proposes XTalker, a controllable, real-time, audio-driven talking portrait animation framework.
It aims to improve beyond conventional parameterized systems that mainly focus on lip synchronization by introducing explicit control over emotion diversity and head motion.

XTalker is built upon a flow-matching diffusion architecture (MM-DiT backbone) and decomposes facial motion into three interpretable subspaces:
	1.	Lip–phoneme synchronization - driven by the temporal dynamics of the audio envelope.
	2.	Emotional diversity - modulated via emotion labels mapped to facial expressions.
	3.	Head motion controllability - achieved through user-defined motion curves and pose parameters.

The model claims to achieve competitive lip-sync accuracy, improved emotion expressivity, and real-time performance (~28 FPS).

**Strengths:**

* The paper is well-structured and addresses a relevant problem: adding controllable emotional and head-motion diversity to speech-driven portrait animation.
* The decomposition into three interpretable subspaces (lip, emotion, pose) is conceptually sound and aligns with the broader goal of human-centered controllable animation.

**Weaknesses:**

1) Missing or overstated claims
	•	The paper claims real-time performance, but no supplemental video, demo, or timing benchmark is provided to verify runtime or visual smoothness.
	•	The statement that the model achieves “precise lip–phoneme synchronization” using amplitude envelope dynamics is overstated.
While the envelope correlates with rhythm and intensity, it does not encode phoneme-level articulatory information.
The observed correlation mainly reflects speech energy, not phonetic content. To substantiate this claim, the authors should report phoneme-level metrics such as SyncNet confidence, viseme accuracy, or phoneme alignment error.

2)  Methodological clarity
	•	The notion of “user-defined head trajectories” is not clearly defined. how are these curves created, parameterized, or constrained? Are they realistic, or manually drawn?
	•	The emotional control relies on mapped labels, but the paper provides little detail on how these emotion embeddings are extracted or how robust they are to noisy or synthetic emotion inputs.

3) Conceptual issues
	•	The audio envelope is treated as a main driver of lip synchronization, but this assumption neglects the contextual and phonetic structure of speech.
The envelope may help timing but cannot enforce accurate articulation.
	•	The claimed decomposition (lip sync, emotion, pose) is presented as a “contribution,” but it is more of a design choice or model architecture feature than a novel methodological advance.

4) Lack of qualitative validation
	•	The paper provides no supplemental videos or qualitative figures showing motion smoothness, expressivity, or emotion diversity.
Given that portrait animation quality is perceptual, the absence of visual evidence significantly weakens the paper’s impact.

**Questions:**

1.	How are user-defined motion curves obtained or parameterized?
2.	Can phoneme-level SyncNet or viseme metrics be reported to validate lip-sync?
3.	Can the authors share runtime measurements or real-time demo videos to support their performance claim?
4.    How would the model generalize to out-of-distribution subjects or in-the-wild data ?

---

> ### Author Response · Authors · 2025-11-13
> **Further Confirmation**
>
> Dear Reviewer sJRq,
>
> Thank you very much for taking the time to review our submission and for providing detailed comments. We noticed that the final rating was “0: strong reject.” Since this score is rarely used in the standard ICLR rating scale, may we kindly confirm whether the score was intended, or if it might have been selected by mistake?
>
> We ask only to ensure that we correctly interpret the evaluation and address all concerns appropriately. Thank you again for your thoughtful feedback.
>
> Authors

---

> > ### Comment · Reviewer_sJRq · 2025-11-13
> > **Confirmation of the review:**
> >
> > Dear Authors,
> >
> > Thanks for writing back. I can confirm that the score I provided was intended.
> >
> > As mentioned in my review and reviewer akiD's comments, evaluating a talking-face generation paper without a supplemental video is extremely challenging. Much of the perceived quality, realism, and temporal consistency of such models can only be meaningfully assessed through visual inspection. In the absence of these materials, it was difficult to fully validate the qualitative claims of the paper, and this substantially impacted my ability to judge the overall contribution.
> >
> > The numerical results and written descriptions alone were not sufficient for me to confidently evaluate the expressiveness, stability, and generalization of the proposed method, which is why the final score reflects this limitation.
> >
> > Thank you again for your submission and for engaging constructively.

---

> > > ### Author Response · Authors · 2025-11-14
> > >
> > > Dear  Reviewer sJRq,
> > >
> > > Thank you for the clarification.
> > >
> > > We sincerely apologize for forgetting to upload the supplemental videos. We have now prepared all video results and will include them in the rebuttal together with our responses to the reviewers’ questions.
> > >
> > > We hope these materials will address the concerns and clarify our method’s qualitative performance.
> > >
> > > Authors

---

> ### Author Response · Authors · 2025-11-28
>
> Dear Reviewer.
>
> We appreciate the reviewers’ efforts in evaluating our submission.
>
> We have updated the work by adding qualitative comparison videos that better demonstrate the method’s performance. We hope the new evidence will support a more complete academic assessment and help clarify earlier concerns.
>
> We will continue revising the paper following the suggestions provided.
>
> Best regards,
>
> Authors

---

### Author Response · Authors · 2025-11-28

Dear All,

We thank the reviewers for pointing out the missing qualitative comparison videos.
In this revision, we have added complete qualitative comparisons, including side-by-side video results that clearly illustrate the visual differences across methods. We kindly invite the reviewers to re-examine the method with these new materials.

It is also important to highlight that our results were obtained under a small-data and small-model training setting, which differs from many recent works that rely on large-scale pretraining or high-capacity generators. Despite these constraints, our model still achieves competitive visual quality, and we believe this demonstrates the effectiveness and efficiency of our framework.

We will release the implementation — including data processing, training scripts, and inference pipeline — after the review period to ensure transparency and reproducibility for the community.

We sincerely hope the newly provided materials support a more balanced and fair evaluation. We do not seek prolonged rebuttal; after reviewers consider the updated evidence, we will respect the final decision accordingly.

Best regards,

Authors

---

### Meta-Review · Area_Chair_WMQ3 · 2025-12-24

**Summary:**

This paper proposes a method called Xtalk, which includes a fundamental approach to controlling the interaction and movement of digital humans in talking portrait animation. A key feature of this paper is its use of a fast and controllable flow-matching method to solve the problem, and experimental results largely validate the paper's initial conclusions.

**Reviewer Concerns:**

+ This paper received relatively negative feedback during the initial review, with reviewer sJRq issuing a very low score of 0. This drew strong attention from the authors and the AC. After several discussions, the authors acknowledged that they had not provided sufficient evidence to prove the validity of the results, such as visualized video results or demo projects.

+ Furthermore, the authors received two explicit negative comments with no indication of improvement: Reviewer akiD indicated limited improvement compared to existing methods, metric errors, etc.

+ Reviewer 7yms stated that the experimental results were insufficient and the innovation limited, citing numerous references for analysis and comparison.

**Reviewer Scores:**

Based on these comments, it can be determined that the current version of this paper received three relatively strong negative comments: 0, 2, and 2 points, with no signs of improvement. The authors have not submitted a rebuttal to address these issues.

---

### Decision · Program_Chairs · 2026-01-26

Reject